# Hospitalization of Transgender Youth in a Psychiatric Ward—Opportunities and Challenges: A Case Study

**Alex Gizunterman** [1,2,*]**, Maya Szczupak** [1]**, Tanya Schechter** [1] **and Yoav Kohn** [1,2] 

1   The Donald Cohen Child and Adolescent Psychiatry Department, Jerusalem Mental Health Center, Eitanim Psychiatric Hospital, 9097200 DN Shimshon, Israel; mayasz84@gmail.com (M.S.); tanya.shechter@psjer.health.gov.il (T.S.); yoavk@ekmd.huji.ac.il (Y.K.)
2   Hadassah School of Medicine, Hebrew University, 9112102 Jerusalem, Israel
\*   Correspondence: gizunterman@gmail.com

**Abstract:** The prevalence of children and adolescents who identify themselves as transgender is significant. Transgender youth are at a high risk for mental health problems, sometimes requiring hospitalization in a psychiatric ward. This situation is specifically complex and should be considered by all mental health professionals. In this case report, we describe the challenges that emerged during hospitalization of a transgender adolescent, followed by descriptions of our attempts to cope with these particular issues.

**Keywords:** gender dysphoria; transgender adolescent; psychiatric institutions

## 1. Introduction

Transgender is a wide term used to describe individuals whose gender identity (one's internal sense of being male, female, or of indeterminate sex) is different from their assigned sex at birth [1,2]. The term cisgender is used to describe individuals whose gender identity is congruent with their assigned sex at birth [3]. Gender dysphoria is defined as marked distress caused by an incongruence between one's gender identity and sex assigned at birth [4].

The prevalence of children and adolescents who identify themselves as transgender is estimated to be between 1% and 3.5% [5]. Because of their deviation from conventional gender norms they are often subject to discrimination, stigma, bullying, and victimization [1,6,7]. The 2011 National School Climate Survey of LGBT (lesbian, gay, bisexual, and transgender) youth [8] screened 8500 students ages 13–20 in the USA. Of the 700 who identified as transgender, 80% reported feeling unsafe at school and more than 50% experienced verbal harassment because of their gender expression. In a study of 8500 students in New Zealand by Clark et al. [9], transgender students were less likely to report protective factors such as family support and were at increased risk for violence compared to their cisgender peers.

In the past decade there have been an increasing number of studies regarding the mental health of transgender youth. These studies have demonstrated that transgender youth have a 2–3-fold risk of depression, anxiety, suicidal ideation, suicide attempts, and non-suicidal self-injury (NSSI) [9–13]. In a retrospective study conducted at a community clinic in Boston [10], transgender youth had a higher probability of being diagnosed with depression and anxiety compared to cisgender controls (50.6% and 26.7% vs. 20.6% and 10%), and had higher rates of suicide attempts (31% vs. 11%), suicidal ideation (56% vs. 20%), and NSSI (30% vs. 8%). Holt et al. [14] examined the difficulties faced by 218 children and adolescents aged 5–17 years who presented to a gender clinic in London during 2012. The most common difficulties reported were bullying (47%), low mood/depression (42%), and self-harm (39%).

The increased risk for mental health problems could explain why a greater proportion of transgender youth access both inpatient and outpatient mental health services, compared with cisgender youth [10].

Various barriers that prevent transgender individuals from accessing medical and mental health treatment have been identified. Lack of provider knowledge about transgender care, lack of sensitivity to transgender needs, discrimination such as using the wrong pronoun or name, or lack of gender-neutral bathrooms in health care facilities are common barriers to care [15–17]. The National Transgender Discrimination Survey found that 50% of patients reported provider ignorance regarding transgender care and 28% postpone medical care because of discrimination [18].

Literature regarding the management of transgender youth in inpatient psychiatric units is scarce, but it appears that inpatient psychiatric units experience challenges similar to other healthcare settings [19].

Here we describe the case of a transgender adolescent admitted to our psychiatric ward that exemplifies challenges and opportunities that may arise in these cases. Information about the patient's past history was gathered from several clinicians treating him prior to admission. Reports on his symptoms and signs at the ward were given by numerous staff members. Follow-up of his condition after discharge was obtained from the treating psychiatrist at the outpatient clinic.

## 2. Patient Description

Bill (pseudonym, some changes in personal details have been made to hide the identity of the patient) is a 17-year-old adolescent who was hospitalized in our department for a period of four months. He identifies himself today as a male, while the sex assigned at birth was female. This was his first hospitalization, and he was treated before by mental health providers only as an outpatient. He was hospitalized in our department because of suicidal ideation, upon his consent and the consent of his parents.

### 2.1. Background

Bill is the oldest of four children. The parents are immigrants to Israel from the United States, both with an academic education and working in their professions. The mother suffers from depression. The pregnancy was described as planned and wanted. Vaginal delivery was at week 38 with no complications. Early motor development was described as normal. However, Bill started talking late and went to a special education kindergarten for children with speech and language problems. He entered a regular first grade class but had a hard time adjusting. Therefore, he was transferred to a small special education class. At that time, he was treated by a psychologist at a mental health clinic because of the existence of an "imaginary friend" and outbursts of anger. As a result, he underwent a psychological evaluation, which identified emotional dysregulation, adjustment difficulties, and anxiety. Psychological treatment helped him, and Bill graduated from elementary school successfully, both academically and socially.

Upon transition to middle school (seventh grade in Israel), there was a significant deterioration in Bill's condition. At the same time, the family moved to another neighborhood, which distanced him from his friends and reduced his social ties. This adversely affected his mood and resulted in a decline in his academic performance. The parents report that during that period Bill began to identify himself as a male. Since then, his mental state has continued to deteriorate. He was preoccupied with sexuality, watching pornographic images and reading pornographic stories with sadistic and pedophilic content. Bill was a member of a WhatsApp group where sexual content was shared. Upon admission to the department, Bill told the staff that during that period he also suffered from sexual harassment. In eighth grade he was sexually harassed by girls at school. In ninth grade a boy harassed him because he was transgender. In tenth grade he was subjected to additional harassment online. Bill reported on suicidal tendencies during this time period. From the age of 13 to the age of 16 he used to cut himself every day.

At the age of 13.5 Bill told his parents that he felt like a boy. The parents did not accept this and continued to talk to him and treat him like a girl. This caused a significant deterioration in the

family relationships, culminating in Bill leaving home and going to a hostel for LGBT teenagers in a different city. For about six months, during which Bill lived in the hostel, his condition improved and his parents were optimistic. Two weeks before he was hospitalized, however, conflicts with the staff and tenants of the hostel began. He had the subjective feeling that all tenants in the hostel were "against him." Bill suffered from outbursts during which he threatened to injure himself and commit suicide. He would slam his head against the wall. He was referred to hospitalization by the staff of the hostel. Until his admission he did not start a process of hormone therapy or gender affirmation/confirmation surgery.

On admission to the department Bill looked according to his age, his hair dyed fluorescent blue with a short haircut. There were significant psychomotor restlessness, crying, screaming, and cursing the staff. His mood was tense and irritated, the affect was appropriate. There were no disturbances in thought process. Thought content consisted of reports of a bad mood, claims that the whole world hated him and that he was on the verge of suicide. There were no signs of delusions or disturbances of perception. Reality testing was normal. His judgment and insight were impaired.

There were discussions among staff members about the differential diagnosis. The working diagnosis was gender dysphoria and depression, and a treatment plan was started accordingly. Other suggested diagnoses included borderline personality disorder, due to the affective instability and externalizing behaviors, and PTSD, given the trauma that he reported.

During hospitalization Bill demonstrated distress to the point of dissociative states. He expressed extreme statements such as "I want to rip my uterus out." He wrapped a shirt around his neck. He had a low frustration threshold. Although Bill repeatedly stated that he did not need hospitalization, he later started cooperating with treatment. He said that he realized he needed help with emotional regulation following mood swings, difficulty sleeping, and a lack of understanding of social situations.

During hospitalization Bill received multidisciplinary treatment. He was given low doses of quetiapine (50 mg/d) as a tranquilizer. During individual psychotherapy and individual art therapy sessions Bill spoke about the sexual harassment that he experienced and about his wish to immediately start cross-sex hormone therapy. He also participated in a DBT (Dialectical Behavioral Therapy) group. His parents received weekly guidance sessions. Gradually, Bill felt more secure and relaxed in his relationship with the staff, and indeed became calmer. He managed to deal better with frustration and not to react with an outburst if his requests were declined.

In the last month of hospitalization, Bill had balanced mood and behavior, with no signs of psychosis or affective disorders, and no abnormal or suicidal behavior. He demonstrated high motivation to move to a residential treatment center (RTC), especially after a visit at the institution. After four months at our department Bill was discharged home and from there he moved to the RTC. Several years after the discharge Bill's mental state is stable, he takes hormonal therapy, and he is going to take part in a national civil service program.

*2.2. Ethical Consideration*

The study was conducted in accordance with the Declaration of Helsinki, and the protocol was approved by the Ethics Committee of Jerusalem Mental Health Center (No. of application in the institutional committee: 19–20).

## 3. Challenges and Opportunities

*3.1. Challenges at the Department*

In our ward (as in most hospitals in our country), bedrooms, toilet rooms, and showers are not private but shared, and are divided into male and female facilities. From the beginning of his hospitalization Bill expressed concern about entering the girls' bathroom and staying overnight in the department. He did not agree to sleep in the same room with girls. When the staff asked him to do so, he reacted with a severe outburst of rage. As a result, it was decided to provide him with a private

room (luckily available at the time), which he filled with flower pots and lots of personal belongings. It seemed as if he was creating a permanent place for himself. Bill also needed a separate shower and toilet because he did not agree to share them with girls, and because the staff felt that letting him share the shower and toilet with boys was impossible, as well. An agreement was made with Bill that he would use the girls' bathroom and the shower room at a certain time, and that at this time the rooms would be used only by him.

### 3.2. Challenges in Contact with the Staff

Bill was one of the first transgender patients hospitalized in the department, and the staff did not have much prior experience in treating this group. Bill went through a significant process in his relationships with the staff. At first, he was severely anxious and experienced any interaction with the staff as aggressive and lacking empathy. He felt that the staff were harassing him and not accepting him. Indeed, sometimes the staff had difficulty addressing him as male rather than as female, especially in typical female contexts, such as his menstrual cramps. Some staff members felt that his behavior was influenced more by personality disorder features than by gender dysphoria. This was dealt with by discussions in departmental staff meetings and a special case presentation to the hospital staff, with a review of the literature on mental health issues unique to transgender youth.

Despite treatment with contraceptive pills and medroxyprogesterone injections Bill still had a menstruation, which worsened his condition. He was screaming that the staff did not care about him and that they wanted to kill him. To address this, he was referred to a follow-up by a gynecologist specialized in the care and treatment of transgender patients.

### 3.3. Challenges in Interpersonal Relationships

Since his admission to the department Bill preferred being addressed by male pronouns. He described doubts regarding the extent to which others accepted and understood him. Some of the other inpatients accepted him very well, but others had difficulties dealing with him. This was especially difficult for ultra-religious boys. His physical condition (Bill talked about his menstruation in public, repeatedly and loudly) confused them greatly. It was hard for them to grasp the concept of "a boy in a girl's body." During the hospitalization, it was difficult for Bill to participate in group therapies and allow discussion of subjects other than transgenderism. He would take the group's "stage" and would not allow talking about any problems of other teenagers. Mediators of group sessions helped Bill improve his self-awareness through the process of interacting with others. Over time, he was able to allow discussions of other subjects, but even in these cases he was immature and extreme in his attitude. The staff's explanations on the issue of gender and transgenders helped the hospitalized adolescents in avoiding a judgmental position, while affecting the staff itself in the same way. Bill created a group of friends in the department. This group demonstrated prominent childish behavior.

### 3.4. Challenges in Treatment

Specialized transgender care is a multidisciplinary approach [20]. Throughout individual therapy sessions, Bill reported feelings and experiences of loneliness. He said that other people harmed him because of his sexual identity. He felt social rejection, despite all his efforts and attempts to become a part of the peer group. Psychotherapy treatment was focused on both gender dysphoria and components of depression. The feelings of rejection were particularly intensified because his parents did not accept him as a male. In discussions with him and his parents they expressed hesitation regarding the appropriate measures to be taken: Should he settle for a hormonal change or undergo gender reassignment surgery, which, according to Bill, may be "complicated and even dangerous"? During hospitalization the patient was in contact with a clinic dedicated to transgender people where he received information regarding hormone therapy and gender affirmation/confirmation surgery. Bill shared his doubts about it with the staff, who discussed the options with him but did not take a position. The staff provided the parents with guidance that helped them discuss these complex

issues and reduce conflicts with Bill. We encouraged Bill and his parents to find common grounds and discuss points of contention.

## 4. Discussion

The prevalence of children and adolescents who identify themselves as transgender is more than 1% of the general population [9,10]. Recent studies have demonstrated that transgender youth are at high risk for mental health problems including suicidality, which could explain the high number of transgender youth who access mental health services [12,13]. Prior research has identified a general lack of access to health services and a lack of continuity of caregiving by families and communities [11]. Because of social rejection and because of the severity of the disorders, sometimes transgender people need to be hospitalized in psychiatric wards.

Our psychiatric ward, like others serving children and adolescents, admits patients with a broad spectrum of mental disorders, coming from varied ethnic and social backgrounds. We make efforts to provide personally tailored treatment plans, taking into account both clinical and social issues. Recently, some transgender youth were hospitalized at our ward, and we found ourselves coping with new challenges. These challenges are not unique to our department as transgender youth represent a population with special needs that have not been fully explored [5].

Through this case study of a transgender adolescent inpatient, we tried to address particular issues and needs that emerged during the period of hospitalization. Some of these issues needed an immediate decision—such as if the youth was going to sleep with girls or boys and which bathroom he or she was going to use. These issues are better discussed with the patient, and the staff should inform him/her about the possibilities that the ward may provide. Our patient often complained of discrimination, even though the staff did not feel that this was the case. We found that involving the patient in decision-making contributed to his/her sense of equality and inclusion.

Another challenge was to discuss the particular characteristics of this youth with other adolescent inpatients and the staff. We found that many of them did not have any knowledge about transgender identities or that they held a judgmental position. Indeed, it has been previously suggested that while schools can be sites of minority stressors, they can also serve as sites for protective factors that enable LGBT youth to thrive in the face of these stressors [21].

It is important to educate the other hospitalized adolescents. In that way we can avoid offensive conflicts and improve the self-esteem and feeling of inclusiveness.

The staff at the hospital ward found themselves in profound debates about Bill's diagnosis and about a therapeutic approach. Routine staff meetings were very important, because they gave us an opportunity to integrate the different observations and to discuss the differential diagnosis of the patient. The identity and behavior of transgender individuals are often socially and medically stigmatized, resulting in an underserved population at risk for negative health outcomes [11]. Gaps in knowledge and biases among medical staff must be determined, and validated tools must then be developed to close those gaps [22].

We noticed that prejudice can bias the concept of diagnosis, but also that increased awareness of transgender needs can make staff members overlook other diagnoses.

The difficulties that we discuss here, like technical limitations of the department and tense interpersonal relationships, are known barriers to accessing primary and mental health care by transgender patients [15–17]. As we and others in psychiatric inpatient wards treat more transgender adolescents, we are expected to expand our professional skills and share our experience with other mental health providers. This is bound to improve the treatment of this special group of patients.

## 5. Conclusions

The hospitalization of transgender youth is becoming part of routine work in psychiatric wards. The logistic and technical structure of the ward, such as providing unisex rooms and bathrooms, as well as educating staff about the specific needs of this population, should be implemented in

all psychiatric institutions. Challenges met in the process are also opportunities for improvement in individually tailoring treatment plans in general. Dedicated funding to ensure consistency of definitions for health surveillance and research initiatives involving transgender people are essential to inform evidence-based decisions [23].

The research literature on this topic is extremely limited. We hope that this article will encourage research in the field of transgender care in a psychiatric ward.

**Author Contributions:** Writing—original draft preparation, A.G., T.S. and M.S.; writing—review and editing, Y.K. All authors have read and agreed to the published version of the manuscript.

**Funding:** This research received no external funding.

**Conflicts of Interest:** The authors declare no conflict of interest.

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
