# Peer review of "Hospitalization of Transgender Youth in a Psychiatric Ward—Opportunities and Challenges: A Case Study"

_sexes, doi:10.3390/sexes1010003_

Round 1

Reviewer 1 Report

I read the article attentively. Basically the article reports on the developments Bill goes through in his battle to find a balance in his gender identity. The article meticulously describes this process and that is an important asset of the article. What I miss though is a specific research question or research questions. Lines 58-59 tell the reader that the article tells him/her/indefinite person the story of Bill but in the academic context of what? What does the article aim at? That we should learn from cases like the case of Bill? Is that all? It seems that the article is basically written for the ward where the authors work itself, as shows from lines 184-186. These say that the ward faces new challenges coping with patients like Bill (whom I would describe as well as a spoiled adolescent who ruthlessly exploits his mixed identity feelings to get what he wants). There is in the discussion hardly any reference to relevant literature and similar cases. The article gives the impression that the ward discovered things of how to work with cases like Bill as if others have not experienced that already earlier. I miss a connection here with, as I said, similar cases or research and experiences. So, the article could benefit from creating a more academic context in which the case of Bill is treated.

Ethics demand that persons mentioned in academic articles cannot traced back after their real identity. Given the fact though that the article mentions the name of the ward where Billy was hospitalized and the fact that so many characteristics of Billy were mentioned, I cannot escape the idea that Billy can be tracked if he reads the article or others close to him. Give this point some more consideration. Also I think some more information must be given as to how his behavior was followed and if the observations done were reliable and/or confirmed by more than one person. Basically then some more feedback on how the data on Billy were collected and how their reliability was established.

Reviewer 2 Report

Overall, this article could be a strong contribution to the literature with some reorganization of the challenges and responses and additional evidence supporting the discussion. I appreciate the opportunity to have read this work. This article would be strengthened by reorganizing the challenges to mirror section 3.2 where authors describe the specific challenges and the response of the team to address these challenges. All sections should include specific actions from the team if they are going to be reported here. Additionally, the authors suggest multiple strategies to improve transgender people’s mental health and healthcare process (e.g., using preferred name, pronouns, changing their bracelets, etc.) but these are not addressed in the challenges/responses section. Please identify whether you used any of these strategies that have been recommended by other transgender individuals in the past. Additionally, there appears to be some new strategies or responses that authors describe in the discussion, but it is unclear as to whether those were used as strategies to address the identified challenges – please see the details in the discussion section and clarify. And finally, please update the discussion to include additional evidence supporting the authors ideas.

Please see below for more detailed comments by section:

  • Title and abstract summarize the overall article well.
  • Background:
    • Overall, the background is a strong summary of the challenges transgender patients face in healthcare settings, their increased risk for co-occurring mental health conditions, and includes a strong definition of transgender.
    • The final paragraph (starting at line 50) repeats some of the same barriers that were identified in other medical care settings in the paragraph prior. Please indicate that although the literature is scarce it appears inpatient psychiatric units experience similar challenges to other healthcare settings to avoid repetition.
  • Patient Description:
    • Please edit line 62 “while being born as a female” to specify while their sex assigned at birth was female sine “sex assigned at birth” is currently the most well accepted terminology per the trans community. (https://transstudent.org/about/definitions/)
    • Please include Bill’s preferred pronouns.
  • Background:
    • Line 98, please consider changing “hormonal or surgical sex change” to “hormone therapy or gender affirmation/confirmation surgery” to be more sensitive to the preferred terminology among the trans community. (https://www.plannedparenthood.org/learn/gender-identity/transgender/transgender-identity-terms-and-labels; https://fenwayhealth.org/documents/the-fenway-institute/handouts/Handout_7-C_Glossary_of_Gender_and_Transgender_Terms__fi.pdf).
    • In the final paragraph of this section you report Bill is taking hormonal therapy. Was this considered as a part of your treatment to address gender dysphoria seeing as it is an evidence-based response? If not, why not? Please address this in either the background or challenges section if there was a specific reason for not considering it (e.g., family concerns or staff biases).
    • This section would be strengthened by organizing your findings and treatment by need, for example the interventions used to address gender dysphoria vs. depression specifically.
    • DBT as an acronym also needs to be defined on line 118.
  • Challenges:
    • Challenges at the department – please elaborate on the rules of your organization/institution or the laws in your area so that readers can better understand why Bill would have been forced to share bedrooms, showers and restrooms with females and not males. Also, I would suggest using caution around the “staff demanding him to do so” as it sounds very aggressive on part of the staff.
    • The organization of 3.2 challenges with staff is very strong. Please consider revising all sections to follow this pattern as it provides the challenge and your specific solutions to address these challenges.
    • Section 3.3 Challenges with interpersonal relationships reads very degrading of Bill. It is unclear why other patients knew of his menstruation – if this is something that he would talk about during groups please clarify that. Perhaps try to restructure this similarly to section 3.2 where you describe his challenging behaviors with interpersonal relationships and then your efforts. Currently this reads as if the only attempt to address this was the mediators of group sessions and this was only partially successful. What other recommendations were provided to address this? Also, the final sentence in this paragraph reads as demeaning to Bill’s success of finding a group of friends with similar interests.
    • Section 3.4 – I wonder if this section should be renamed and focus specifically on family interventions? This section also needs more information about what guidance the staff provided parents with to help reduce conflicts with Bill and more information about the impact of these interventions on the family unit.
  • Discussion:
    • Please avoid suggesting someone’s experiences of discrimination are exaggerated, even if you feel that they may be. Instead, please focus more on the inclusion of the patient in their care to provide autonomy. There is other work that you could cite that would support this statement in general for healthcare processes that would strengthen your discussion section.
    • The paragraph beginning on line 194 is particularly interesting, I would have liked to see this come up in your challenges/responses section if this is something you did. Also, to strengthen the discussion you could pull existing data that supports outcomes for all youth (especially mental health outcomes for transgender youth) when others are educated (Kosciw et al., 2012).
    • The paragraph on 199 is particularly interesting and this discussion section would be strengthened if you expanded upon these ideas with relevant citations supporting them.

Round 2

Reviewer 1 Report

Better now. I read the comments of the authors on my comments and find them reasonable and good that they changed the identity of "Billy" more so that he becomes less recognizable although there remains the possibility that people will recognize him. But authors did their best. Also their remarks that it is more the presentation of a case than an academic study is okay. I see that back in the new version of the paper as well.